# A Self-Healing Ionic Liquid-Based Ionically Cross-Linked Gel Polymer Electrolyte for Electrochromic Devices

**DOI:** 10.3390/polym13050742

**Published:** 2021-02-27

**Authors:** Wanyu Chen, Siyuan Liu, Le Guo, Guixia Zhang, Heng Zhang, Meng Cao, Lili Wu, Tianxing Xiang, Yong Peng

**Affiliations:** 1School of Materials Science and Engineering, Wuhan University of Technology, Wuhan 430070, China; chenwanyu@whut.edu.cn (W.C.); liusiyuan_YSL@whut.edu.cn (S.L.); zhang94670@whut.edu.cn (G.Z.); zhang-hg@whut.edu.cn (H.Z.); caomeng19970705@163.com (M.C.); polymer_wl@whut.edu.cn (L.W.); 2State Key Laboratory of Advanced Technology for Materials Synthesis and Processing, Wuhan University of Technology, Wuhan 430070, China; catchyue@gmail.com (L.G.); xiangtianxing@whut.edu.cn (T.X.)

**Keywords:** electrochromic device, ionically cross-linked gel, electrolyte, ionic liquids

## Abstract

An ionic liquid-based ionically cross-linked gel polymer electrolyte (GPE-ILs) was successfully synthesized using acrylic acid, 2-diethylaminoethyl methacrylate, methyl methacrylate, and ionic liquids. Electrochromic devices (ECDs) with an architecture of glass/FTO/WO_3_/GPE-ILs/FTO/glass were fabricated by a laminating technology. The devices showed performances of large optical modulation of 49.9% at 650 nm, short switching times with the coloration time (tc) of 7 s and the bleaching time (tb) of 4 s, high coloration efficiency of 96.2 cm^2^ C^−1^, and cycling stability of 200 cycles. The GPE-ILs exhibits high ionic conductivity, superior thermal stability and good self-healing ability. GPE-ILs demonstrates an ionic conductivity of 3.19 × 10^−3^ S cm^−1^ at 25 °C and the same ions migration behaviors with most widely used liquid electrolyte between −10 and 80 °C maintains more than 80% of its tensile strength after self-healing and received only 5% weight loss at 300 °C.

## 1. Introduction

Electrochromic devices (ECDs) have attracted extensive attentions in last decades because of their applications in energy saving technologies, like smart windows, low-power displays, electronic skins, etc. [1,2,3,4]. However, most ECDs are suffering stability and safety problems even they have advantages of low power consumption, high color rendering efficiency, short switching time, etc. [5,6,7,8]. 

In a typical ECD device, electrolytes with high ionic conductivity, high transparency, good thermal stability, and good electrochemical stability were desired [9,10,11,12]. Liquid electrolytes and solid electrolytes are popular, nevertheless, liquid electrolytes are encountering volatilization and leakage [13,14] and solid electrolytes are suffering from low response time and non-uniform coloring [15,16]. Gel polymer electrolytes (GPEs) were considered as one of the ideal electrolyte candidates because of good mechanical properties of molding ability (e.g., character of self-healing) [17,18,19] but ion conductivity at subzero temperatures and fatigue properties need a further improvement before considering real applications [20,21].

Non-covalent bonds (physically cross-linked) and dynamic covalent bonds (chemically cross-linked) are strategies of synthesizing self-healing gels [22], self-healing materials can heal themselves after suffering external mechanical damage or harsh environments, thereby effectively extending the service life of the material and improving its safety [23]. Generally, non-covalent bonds such as hydrogen bonding, hydrophobic bonding, and electrostatic attraction can be used as the driving force for the self-healing of materials [24,25,26]. The ionically cross-linked polymer network is combined by the electrostatic attraction of high density between negatively and positively charged groups on polymer chains [27,28]. When the ionically cross-linked polymer network is destroyed, due to the movement, rearrangement, and electrostatic attraction of the polymer chains, the network reconnects to form ionic bonds that promote the self-healing of the damaged area [22]. Ionic liquids (ILs) have many excellent properties, such as low vapor pressure, nonflammability, high thermal stability, high ionic conductivity, and wide electrochemical window [29,30,31,32]. GPEs will receive an improved ionic conductivity and long-term stability if ILs were introduced, e.g., Fernandes et al. used sol–gel method to prepare an organic–inorganic hybrid material containing ionic liquids as an electrolyte in ECD [33,34]. Studies about ILs-based chemically cross-linked GPEs have been commenced [35,36,37,38] but only a few researchers synthesized transparent ionically cross-linked GPEs because of compatibility between ILs and ionically cross-linked networks. For applications of ECDs, a transparent GPE was expected. Therefore, in order to obtain ILs-based ionically cross-linked GPEs with high transparency, the selection of ILs and polymer monomers is particularly important. Here, we compare the compatibility of the three ILs with different polymers. The polymer monomers consist of acrylic acid, 2-diethylaminoethyl methacrylate, and six types of olefin monomers (Appendix A).

The conductivity, thermal stability, and optical properties of GPEs were obviously improved by introducing an 3D ionically cross-linked network trapped IL-1-ethyl-3-methylimidazolium bis(trifluoromethylsulfonyl)imide ([Emim]TFSI) and a highly conductive Li+ source LiTFIS into GPE (GPE-ILs). Switching speed, optical modulations, and cycling stability of ECDs adopting this GPE-ILs were investigated afterwards.

## 2. Materials and Methods

### 2.1. Materials

Methyl methacrylate (MMA, anhydrous, 99%), acrylic acid (AA, purity > 99%), 2-diethylaminoethyl methacrylate (DEA, 99%), ethyl alcohol (anhydrous, 99.7%), and tungsten chloride (WCl_6_, 99.5%) were purchased from Aladdin (Shanghai, China). Moreover, 1-ethyl-3-methylimidazoliumbis[(trifluoromethyl)sulfonyl]imide([Emim]TFSI, purity > 99%) and lithium bis(trifluoromethylsulfonyl)imide (LiTFSI, anhydrous, 99%) were purchased from MonILs Chemical (Shanghai) Co. Ltd., China. Dibenzoyl peroxide (BPO, AR) was provided by Sigma-Aldrich. Platinum electrodes (DJS-1C) were obtained from Shanghai Leici. The FTO-coated glass sheet (sheet resistance: 5–7 Ω sq^−1^) was obtained from Zhuhai Kaivo Optoelectronic Technology Co., Ltd. (Zhuhai, China).

### 2.2. Preparation of the GPE-ILs

Ionically cross-linked gel networks were formed by copolymerization of MMA, DEA, and AA. The synthesis was completed via following steps: 1. 1 mL DEA and 1 mL AA were added into 8 mL MMA solution in order to form a mixture of MDA. 2. Ionic liquid [Emim]TFSI and MDA were mixed uniformly with a volume ratio of 2:1. 3. Then, 10 mg BPO, serving as an initiator, and a certain amount of LiTFSI was added to the [Emim]TFSI and MDA mixture. 4. This mixture were ultrasonicated for 1.5 h until the mixture changes to a clear and transparent solution. 5. The solution was bubbled with nitrogen for 30 min to ensure an O_2_-free environment, and then, the remaining gas dissolved in the system was exhausted. 6. This O_2_-free solution was transferred to a sealed mold composed of two pieces of FTO glass. 7. This mold was then heat treated at 80 °C for 12 h (Figure 1a). The major reaction can be illustrated as two steps: protonation process of DEA and AA and the process of free radical bulk polymerization (Figure 1b). Finally, a free-standing GPE-ILs film was obtained.

### 2.3. Fabrication of the Electrochromic Layers

WO_3_ films were deposited on FTO glasses through a templated sol–gel process in an ultrasonic sprayer (Sono-Tek, Corporation, Milton, NY, USA), which was reported previously [39].

### 2.4. Assembly of EC Device

ECDs with a sandwich structure of glass/FTO/WO_3_/GPEs-ILs/FTO/glass were assembled by adding GPE-ILs in between a WO_3_-coated FTO glass and a bare FTO glass, which is as shown in Figure 2.

### 2.5. Characterization

#### 2.5.1. The Thermal Properties of GPE-ILs

Thermogravimetric analysis (TGA) of GPE-ILs was performed with a Netzsch Instruments simultaneous thermal analyzer (STA449F3) from 25 to 600 °C with a heating rate of 10 °C min^−1^ under nitrogen atmosphere. The differential scanning calorimetry (DSC) of the samples was carried out using a TA-DSC2500 instrument with a temperature ranging from −80 to 150 °C and heating/cooling rate of 10 °C min^−1^. Tensile stress strain curve of GPE-ILs was obtained using an Electronic Universal Material Testing Machine (Instron 5967). The length and diameter of the samples were 40 and 4 mm, respectively, and the stretching rate was 10 mm min^−1^.

#### 2.5.2. The Electrochemical Performance of GPE-ILs

The ionic conductivity of the GPE-ILs from −10 °C to 80 °C were measured by alternating current impedance spectroscopy, which was obtained form an electrochemical workstation (Bio-Logic SP-300) under working conditions of a test frequency range from 0.1 Hz to 1 MHz with the amplitude of 10 mV. The platinum electrodes (DJS-1C) are placed in the unpolymerized precursor solution. After the polymerization is completed, the GPE-ILs can fully contact the electrode, which obviously reduces the test error caused by the poor contact between the gel and the electrode. The electrochemical stability window (ESW) of GPE-ILs was measured by linear sweep voltammetry with a scan rate of 10 mV s^−1^ and a voltage range from 0 to 5 V. The GPE-ILs was sandwiched between two sheets of FTO glass, which acted as the reference and working electrode, respectively.

#### 2.5.3. The Optical Transmittances of the ECDs

Optical transmittances of the ECDs were measured using an ultraviolet visible near-infrared spectrophotometer (Lambda 750S, PerkinElmer) with wavelength ranging from 350 to 800 nm. Electrochemical workstation (Bio-Logic SP-300) was used to measure chronoamperometry (CA) of the ECDs and the applied voltage was stepped between −3.0 and +3.0 V with step intervals of 40 s.

## 3. Results and Discussion

### 3.1. The Thermal Properties of GPE-ILs 

Figure 3a shows DSC thermograms for GPE-ILs over the temperature range from −100 to 150 °C. As shown in the enlarged view of area for glass transition Figure 3b, the starting point temperature of the glass transition is −55.6 °C, and the ending point temperature is −51.2 °C, so the glass transition temperature is −53.4 °C, which is consistent with the above analysis results of GPE-ILs that have no phase transition at −10 °C. When the temperature is higher than glass transition temperature, lithium ions can easily and quickly migrate in the electrolyte, which results in high ionic conductivity [40]. Consequently, the conductivity of GPE-ILs at −10 °C still reaches 5.68 × 10^−4^ S cm^−1^, and when the temperature is above the glass transition temperature of −54.3 °C, the conductivity of GPE-ILs does not decrease sharply due to the phase transition. Therefore, ECDs with GPE-LIs as electrolyte can be used normally in a wide temperature range.

Weight loss of most organic gel polymer electrolytes at 300 °C is more than 15% [12,41,42,43,44] that can cause a serious long-term stability issues. Existence of carbonate-based plasticizers in electrolyte was considered as a major reason. Replacing solvents such as PE and EC with thermally stable [Emim]TFSI can improve thermal stabilities. Figure 3c,d shows the TGA and DTG curves of [Emin]TFSI and GPE-ILs; the TGA curves illustrates that [Emim]TFSI has almost no weight loss at 350 °C. while GPE-ILs based on [Emim]TFSI has a weight loss of 5% at 300 °C probably arisen from volatilization of incompletely polymerized monomers. As being illustrated from TG results, the weight loss stage of GPE-ILs at 300–500 °C is separated into two stages: from 300 to 420 °C and from 400 to 500 °C. At 300–420 °C, the weight loss rate of GPE-ILs is higher than [Emim]TFSI, which indicated that the polymer host of GPE-ILs is decomposed. At 420–600 °C, the weight loss rate of GPE-ILs was basically equal to that of [Emim]TFSI, indicating that the weight loss within this temperature range was accounted for [Emim]TFSI [45]. In general, GPE-ILs based on [Emim]TFSI demonstrated a good thermal stability below 300 °C.

### 3.2. The Ionic Conductivity of GPE-ILs

Figure 4a shows the AC impedance spectrum of [Emim]TFSI and GPE-ILs at 25 °C, an equivalent circuit as shown in Figure 4b is applied to fit experimental results. Therein, *Rs* represents the bulk resistance; Rf is the interfacial resistance; CPE is the constant phase element. In addition, then the bulk resistance values are fitted by Zview software. The ionic conductivity of the electrolytes was calculated according to Equation (1):(1)$$\sigma =\frac{1}{RS}\frac{l}{S},$$
where *σ* is the conductivity, *Rs* is the bulk resistance, *l* is the distance between two platinum electrodes, and *S* is the interface area between the platinum electrode and the PADA gel electrolyte. In the present measurement, *l*/*S* = 2 cm^−1^.

Figure 4c shows the ion conductivity of GPE-ILs and [Emim]TFIS from −10 to 80 °C. The ion conductivity of GPE-ILs is 3.29 × 10^−3^ S cm^−1^ at 25 °C. In comparison, according to numerous reported studies, the GPEs with chemical cross-linked and organic solvents as liquid electrolyte exhibit relatively low ionic conductivities ranging from 10^−5^ to about 10^−4^ S cm^−1^ at 25 °C (Table 1). Figure 4d exhibits the temperature dependence of the ionic conductivity of the GPE-ILs and the ionic liquid. Generally, the relationship between the ionic conductivity of the liquid electrolyte and the temperature can be described by the Arrhenius equation (Equation (2))
(2)$$\sigma =A\exp(\frac{-Ea}{KT}),$$
where *σ* is the ionic conductivity, *A* is the pre-exponential factor, *Ea* is the activation energy, *K* is the Boltzmann constant, and *T* is the temperature in Kelvin.

As shown in Figure 4d, the log *σ* is linear with 1000/T for the GPE-ILs, and the linear correlation parameter r^2^ is 0.995. Therefore, the ions transfer resistance of GPE-ILs is controlled by the thermal movement of ions so that the conductive behavior of GPE-ILs is similar to that of liquid electrolytes [27]. This structure enables the ionic liquid, as a continuous phase was encapsulated in the three-dimensional network of GPE-ILs, forming continuous transport channels to facilitate the diffusion and migration of ions [24]. In particular, at sub-zero temperature, the conductive behavior of GPE-ILs is still similar to that of ionic liquid, which proved that phase transition does not occur and its operability under a low temperature of −10 °C.

### 3.3. The Analysis of the ECD

In our devices, the GPE-ILs as the ion transport layer actually also plays a role in ion storage, and the optical properties of ECDs can be changed because of the insertion/extraction processes of Li^+^ and electrons in/out of the WO3 films. Figure 5a shows the bleaching and coloring state of ECD under a voltage bias of ±3.0 V. The ECD demonstrated a uniform blue color at colored state and transparency at bleached state. As shown in Figure 5b, the transmittance of the ECD in the bleaching and coloring state are 78.6% and 28.8%, respectively. Transmittance variation of the ECD is 49.9% at 650 nm. The transmittance of the ECD at 650 nm is measured in situ with a periodic voltage (depicted in Figure 5c). As shown in Figure 5c, the coloration time (tc) and bleaching time (tb) of ECD are 7 and 4 s, respectively. The switching times are shorter than the previously reported publications (Table 1) due to the high ionic conductivity of GPE-ILs.

### 3.4. The Coloration Efficiency and Cycling Durability of ECD 

Coloration efficiency (*CE*) an important performance of ECDs is defined as the optical change caused by a unit charge density at a given wavelength, which could be determined by Equation (3):(3)$$CE=\frac{\Delta OD}{\Delta Q}=\frac{\log(Tb/Tc)}{\Delta Q},$$
where Δ*OD* represents the change of optical density. *Tb* and *Tc* are the transmittance at the bleached and colored states, respectively, and Δ*Q* can be calculated by the current integration (in Figure 5d). The *CE* value of ECD is 96.2 cm^2^ C^−1^, which is higher than some reported works (Table 1). The results show that ECDs based on GPE-ILs can obtain large optical modulation under low energy input, which is of great significance for energy saving.

A wide electrochemical window of electrolyte will ensure that ECDs can operate stably within the working voltage. Figure 6a shows the result of a linear sweep voltammetry on GPE-ILs and the electrochemical stability window of GPE-ILs is 3.75 V, which indicates that GPE-ILs is stable enough under the normal working voltage (3 V) of ECD. Figure 6b shows the repeated coloring and bleaching process of GPE-ILs-based ECD during 50 cycles. The detail of transmittance for 200 cycles can be obtained in Appendix A. At the beginning, the transmittances of bleached and colored ECD are 77.6% and 27.7%, respectively. *Tb* slightly drops and *Tc* rises after 100 cycles, *Tb* drops to 74.4%, *Tc* rises slightly to 32.3%, and the difference reduces to 42.1%, which represents only a 7.8% decrease. As the cycle number increases to 200, the optical modulation of ECD can be still maintained to 35%. As the cycle number increases, *Tb* gradually decreases, while *Tc* gradually increases, which is due to the ion trapping in tungsten oxide.

### 3.5. The Self-Healing Ability of GPE-ILs

GPE-ILs with good mechanical properties can further improve stability of ECDs. Figure 7b shows in the tensile state and stress-strain curve of GPE-ILs that the breaking elongation of GPE-ILs is 400% and the breaking strength is 45 KPa. In Figure 7c, the GPE-ILs is cut into two parts, and the new section is put together at room temperature (without adding any chemicals) for 12 h. As shown in Figure 7d, the elongation at break of GPE-ILs after self-healing is 340%, and the breaking strength is 35 KPa indicating that the healing GPE-ILs can still maintain a good elongation and a certain strength. There are two main reasons for the self-healing of GPE-ILs. The first reason is when the ionically cross-linked polymer network is cut and placed together again, the network reconnects to form ionic bonds due to electrostatic attraction. On the other hand, the ionic liquid in GPE-ILs promotes the movement of polymer chains, and with the rearrangement of polymer chains, it is more conducive to the interaction of electrostatic attraction in the damaged area [22].

## 4. Conclusions

A self-healing GPE-ILs was successfully fabricated, and ECDs adopting GPE-ILs as electrolyte demonstrated desirable properties of both operating and stabilities. The prepared ECD exhibits a large optical modulation of 49.9% at 650 nm, short response time during the colored (7 s) and bleached (4 s) processes, high coloration efficiency of 96.2 cm^2^ C^−1^, and good cycling performance of more than 200 cycles. These properties improvement can be attributed to the excellent properties of GPE-ILs. The ionic conductivities of GPE-ILs are 3.29 × 10^−3^ and 0.58 × 10^−3^ S cm^−1^ at 25 °C and −10 °C, respectively. The weight loss of GPE-ILs is only approximate 5% under 300 °C, and the glass transition temperature is −53.4 °C. In addition, GPE-ILs has superior self-healing ability that the tensile properties of GPE-ILs after self-healing can still maintain more than 80%. In general, GPE-ILs not only have great potential in traditional electrochromic devices but also are expected to be applied to some new electronic devices such as foldable smart windows and stretchable electronic skins.

## Figures and Tables

**Figure 1 polymers-13-00742-f001:**
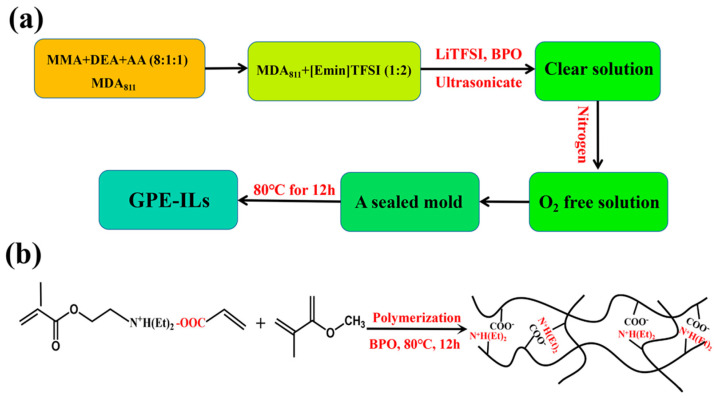
(**a**,**b**) Synthesis process of an ionic liquid-based ionically cross-linked gel polymer electrolyte (GPE-ILs).

**Figure 2 polymers-13-00742-f002:**
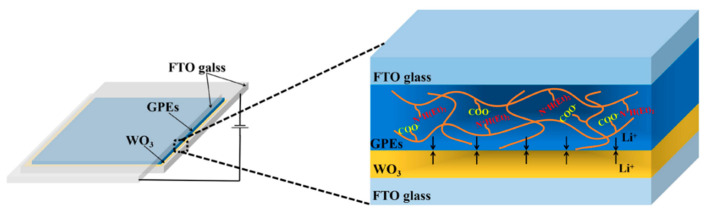
The assembly process of the electrochromic device (ECD).

**Figure 3 polymers-13-00742-f003:**
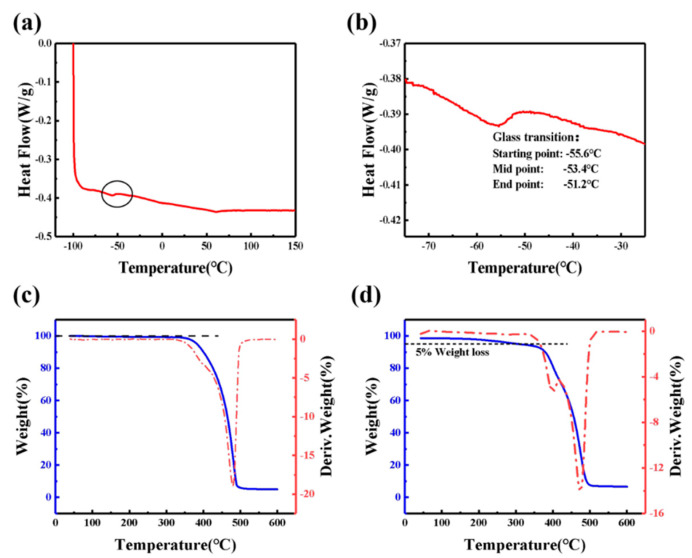
(**a**) Differential scanning calorimetry (DSC) thermograms for GPE-ILs over the temperature range from −100 to 150 °C at a heating rate of 10 °C min^−1^. (**b**) Enlarged view of area for glass transition. TGA and its DTG curve of (**c**) [Emim]TFSI and (**d**) GPE-ILs.

**Figure 4 polymers-13-00742-f004:**
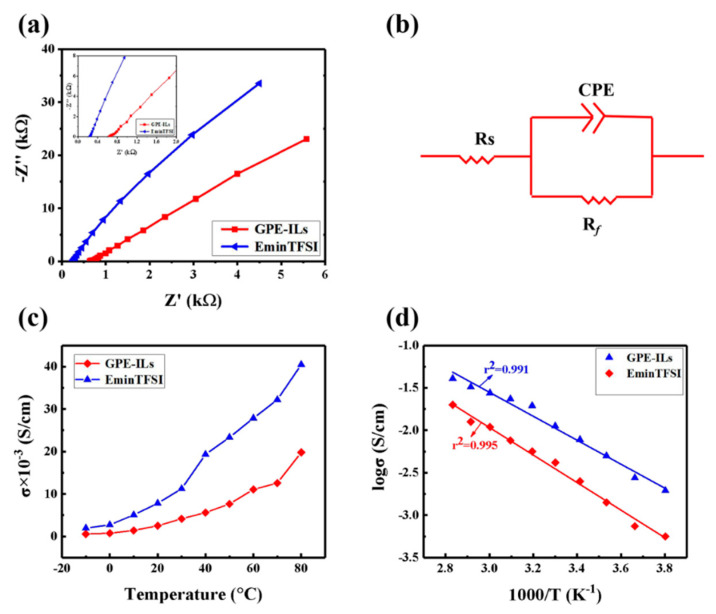
(**a**) Nyquist plots of different electrolytes at 25 °C. (**b**) The equivalent circuit for electrolytes in present work. (**c**) Ionic conductivity of GPE-ILs and the ionic liquid in the range from −10 to 80 °C. (**d**) Temperature dependence of the ionic conductivity of the GPE-ILs and the ionic liquid.

**Figure 5 polymers-13-00742-f005:**
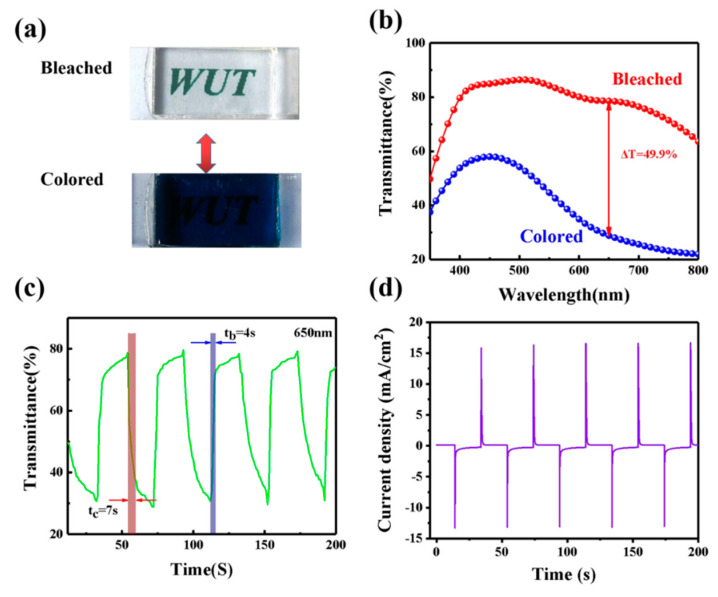
(**a**) Photographs of ECD in bleached and colored states. (**b**) Transmittance spectra of ECD with GPE-ILs in bleached and colored states from 350 to 800 nm. (**c**) Switching speed of the ECD with GPE-ILs. (**d**) Chronoamperometry curve during the switching test. Potential: ± 3.0 V, cycling time: 40 s.

**Figure 6 polymers-13-00742-f006:**
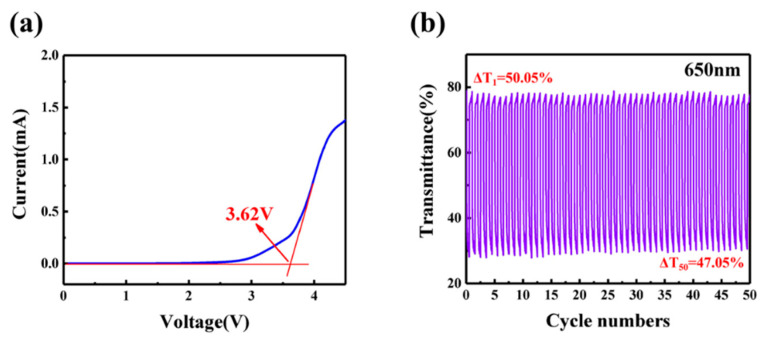
(**a**) Linear sweep voltammetry curve for GPE-ILs. (**b**) Transmittance of ECD with GPE-ILs at 650 nm for 50 cycles.

**Figure 7 polymers-13-00742-f007:**
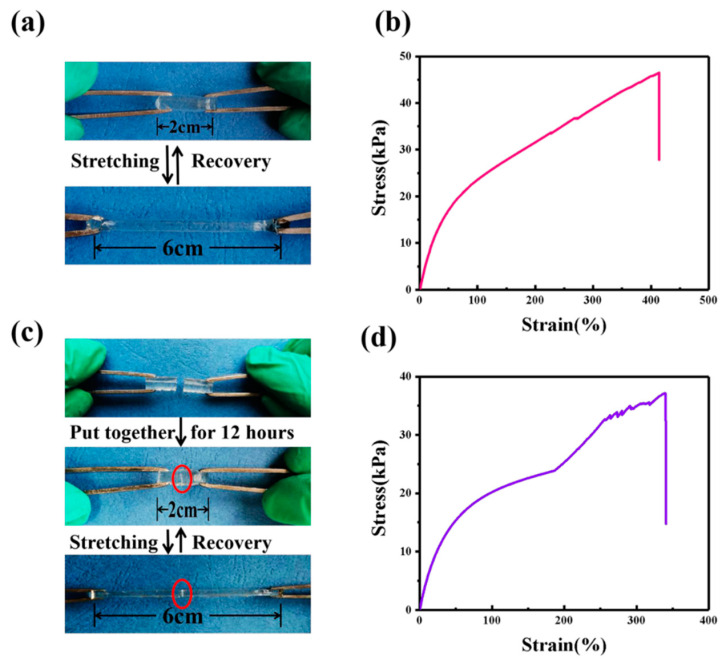
(**a**) Photographs demonstrating stretching of GPE-ILs. (**b**) Tensile stress strain of GPE-ILs. (**c**) Photographs demonstrating self-healing and stretching of GPE-ILs. (**d**) Tensile stress strain of self-healing GPE-ILs.

**Table 1 polymers-13-00742-t001:** Switching times of several reported ECDs and the ionic conductivity of gel polymer electrolyte.

ECDs Construction	Ionic Conductivity (S/cm)	Switching Time (s) tc/tb	Coloration Efficiency (cm^2^ C^−1^)	Ref.
FTO/WO_3_/protonic gelatin-based solid electrolyte/NiO/FTO	1.28 × 10^−^^5^ (25 °C)	30/30	38.1	12
ITO/WO_3_/PVB-based GPEFs/Ni_1−x_O/ITO	4.0 × 10^−5^ (25 °C)	16.5/9.5	175.3	20
FTO/WO_3_/PMMA-[Emim]BF_4_/LiClO_4_/FTO	2.9 × 10^−3^ (25 °C)	62.6/41.2	55.3	21
FTO/WO_3_/PADA gel electrolyte/NiO/FTO	1.33 × 10^−2^ (25 °C)	7.5/8.5	78.7	28
ITO/WO_3_/d-PCL(530)/siloxane_2_[Emim]BF_4_/LiCF_3_SO_3_/ITO	4.0 × 10^−4^ (36 °C)	30/50	152	34
ITO/PANI:DBSA/PVdFHFP-ICPTES-ZrO_2_ /PEDOT:PSS/ITO	2.5 × 10^−4^ (25 °C)	10.06/9.50	/	46
FTO/WO_3_/GPE-ILs/FTO	3.29 × 10^−3^ (25 °C)	7/4	96.2	This paper

## Data Availability

The data presented in this study are available on request from the corresponding author.

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
