# Peer review of "A Self-Healing Ionic Liquid-Based Ionically Cross-Linked Gel Polymer Electrolyte for Electrochromic Devices"

_polymers, 2021, doi:10.3390/polym13050742_

Round 1
Reviewer 1 Report
The structure of the paper shoul de reformulated. And Shoul be clearly indicated wich IL is employed at the current work.
Intoduction
The introduction section is concise. I sugest to the the authors to give a briedf explanation about self-healing and some examples of ECDs.
In the reference [31] it is indicated an organic-inorganic hybrid materials prepared by the sol-gel method that acts as biohybrid electrolyte in the ECD. I suggest to the authors that indicate this and also can include the following reference, from the same group, with different host hybrid but incorporating also ILs and lithium salt: “Electrochromic Device Composed of a Di-Urethanesil Electrolyte Incorporating Lithium Triflate and 1-Butyl-3-Methylimidazolium Chloride".Frontiers in Materials 7 (2020) DOI. 10.3389/fmats.2020.00139
Materials and methods
Both forms are presented in the text: [Emim]TFSI) and EmimTFSI. Please write always in the same way.
The authors said: “…we compare the compatibility of the three ILs with different polymers.”
Could you please explain better the experimental procedure. Did you use the three ILs? In wich quantity employed? How do you distinguished along the text?
Point 6 of experimental procedure , why the FTO plates?
Maybe the authors can do a fluxogram..
Synthesis and deposition of the WO3 EC layer. What about the corresponding characterization?
The assembly of the ECDs is described. Which IL was employed? The three indicated or just one?
Characterization
The description of the characterization techniques is confused. Why it is not included in the previous section of methods?
The characterization of the ECD is presented between the ionic conductivity and thermal properties of GPE-ILs. It is not logical in my opinion.
Page 3, line 111: “The thermal properties of …” should indicate DSC, since TGA already described is also thermal properties in the same way.
Results and Discussion
For me it does not make any sense to present first the ECDs and after the GPE-Ils that give origin to the ECDs. I think the authors prepared the GPE-Ils and chosed the best one to prepare the ECDs? Or used all the GPE-Ils? I really do not understand. The authors should move all the characterization part of the GPE-ILs before the characterization of rhe ECDs.
For instance, the authors present a table 1 with the ionic conductivity of the current work (which system, wich concentration of ILs, wichIL?) and just after this point they present the discussion of ionic conductivity.
Table 1. Please confirm the nomenclature employed in reference 31. d-PCL(530) .
Line 152: Please confirm the way to write 10-5 Scm-1
Line 161-162: Reference to support the sentence is missing.
Line 196-197: Reference to support the sentence is missing.
Line 202: Please indicate the Tb an Tc values and the wavelength or wavelength range in which were determined. How the authors determine the Q value?
Please also include a comment about the DT value (presented in Figure 3)
Line 205: The authors compare the values obtained for CE in the present wok with the CE values found in the literature. Please include in Table 1 of the current work and compare also with the references 21, 25 and 42 and if it is possible, with the references of table 1. For instance the reference 31 also present CE values.
Supporting information
Please define PECD.
In my opinion the Table S1 it is not relevant. Specally when they do not present important information such as quantities employes in the different synthesis. May be the table could be completed with this information.
The authors can present a picture of transparent film to illustrate.
Author Response
We sincerely thank you all for spending precious time and efforts in reviewing this manuscript and greatly appreciating your insightful comments to make the paper better. The manuscript has been carefully revised, improved and verified to address the questions raised by the reviewers. Revisions have been highlighted in yellow for clarification. Below we list responses in the sequence in which they were raised in the referee’s report.

Reviewer 2 Report
This manuscript describes a very interesting subject on the gel polymer electrolyte for electrochromic device. The study includes ionic conductivity, transmission, efficiency and time of bleach and coloration, cyclic stability and tensile strength after self-healing. The cyclic stability is still not long enough for practical applications, nevertheless, it shed a light of using this kind of gel polymer as electrochromic device. Although part of the work follows the methods of the authors’ previous publication (Journal of Material Chemistry in 2019) on PADA ionic cross linking polymer, in this work, the self-healing part is quite special and of useful for flexible substrates. The manuscript should be able to be published on this journal if the following items are fixed.
- What kind of ions transport in the electrolyte and the ion trap in tungsten trioxide? Because there is no layer for ion storage(see Fig. 1), so, where is the ion storage in this device? These parts should be stated clearly in the text.
- The first paragraph in Introduction (line 26 & 28 of page 1): the “et al” is not a right wording. This word is only for persons.
- In line 100 of page 3: The ESW is not defined.
- In line 187 and 188 of page 7: the sequence of “GPE-ILs and Emin[TFSI]” seems revised.
- In line 208 of page 8: The definition of Delta-OD, was defined as the “optical density of the ECD”, which is wrong.
- Line 143 of page 5: Two platinum electrodes is used. It is different from the Fig. 1 for the device (by FTO electrodes). If this is a special arrangement just for electrolyte measurements, it should be stated clearly in the text.
- Line 144 of page 5: There should be unit for l/S = 2, e. g., cm-1.
- There should be a blank space between all the numbers and units.
Author Response

(The authors gave the same response as above.)
